

# Effects of Gas-Wall Interactions on Measurements of Semivolatile Compounds and Small Polar Molecules

Xiaoxi Liu,[1] Benjamin Deming,[1] Demetrios Pagonis,[1] Douglas A. Day,[1] Brett B. Palm,[2] Ranajit Talukdar,[1] James M. Roberts,[3] Patrick R. Veres,[3] Jordan E. Krechmer,[4] Joel A. Thornton,[2] Joost A. de Gouw,[1] Paul J. Ziemann,[1] and Jose L. Jimenez[1]

[1] *Department of Chemistry and Cooperative Institute for Research in Environmental Sciences, University of Colorado, Boulder, CO, USA*
[2] *Department of Atmospheric Sciences, University of Washington, Seattle, WA, USA*
[3] *NOAA Chemical Sciences Division, Earth Systems Research Laboratory, Boulder, CO, USA*
[4] *Center for Aerosol and Cloud Chemistry, Aerodyne Research Inc., Billerica, MA, USA*
Correspondence to: Jose L. Jimenez (jose.jimenez@colorado.edu)

## Abstract

Recent work has quantified the delay times in measurements of volatile organic compounds (VOCs) caused by the partitioning between the gas phase and the surfaces of the inlet tubing and instrument itself. In this study we quantify wall partitioning effects on time responses and transmission of multi-functional, semivolatile and intermediate-volatility organic compounds (S/IVOCs) with saturation concentrations ($C^*$) between $10^0$ and $10^4$ µg m$^{-3}$. The instrument delays of several chemical ionization mass spectrometer (CIMS) instruments increase with decreasing $C^*$, ranging from seconds to tens of minutes, except for the NO$_3^-$-CIMS where it is always on the order of seconds. Six different tubing materials were tested. Teflon, including PFA, FEP, and conductive PFA, performs better than metals and Nafion in terms of both delay time and transmission efficiency. Analogous to instrument responses, tubing delays increase as $C^*$ decreases, from less than a minute to > 100 min. The delays caused by Teflon tubing vs. $C^*$ can be modeled using the simple chromatography model of Pagonis et al. (2017). The model can be used to estimate the equivalent absorbing mass concentration ($C_w$) of each material, and to estimate delays under different flow rates and tubing dimensions. We also include time delay measurements from a series of small polar organic and inorganic analytes in PFA tubing measured by CIMS. Small polar molecules behave differently than larger organic ones, with their delays being predicted by their Henry's law constants instead of their $C^*$, suggesting the dominance of partitioning to small amounts of water on sampling surfaces as a result of their polarity and acidity properties. PFA tubing has the best performance for gas-only sampling, while conductive PFA appears very promising for sampling S/IVOCs and particles simultaneously. The observed delays and low transmission both affect the quality of gas quantification, especially when no direct calibration is available. Improvements in sampling and instrument response are needed for fast atmospheric measurements of a wide range of S/IVOCs (e.g., by aircraft or for eddy covariance). These methods and results are also useful for more general characterization of surface/gas interactions.



# 1 Introduction

Tubing that transports air from the ambient atmosphere or laboratory experiments to a detector can perturb the concentrations of gaseous analytes in the air by gas-wall interactions, and thus presents a challenge to accurate quantification. Teflon is a commonly used material for tubing and analytical instrumentation as well as for laboratory chemical reactors. In recent years, researchers have observed that semivolatile organic compounds partition reversibly to Teflon chamber walls made of fluorinated ethylene propylene (FEP) with typical equilibration time scales of tens of minutes (Matsunaga and Ziemann, 2010; Yeh and Ziemann, 2015; Krechmer et al., 2016; Ye et al., 2016; Huang et al., 2018). Similarly, the same type of partitioning also occurs with inlet tubing and instrument surfaces. Pagonis et al. (2017) systematically investigated the delays of volatile organic compounds (VOCs) and intermediate volatility organic compounds (IVOCs) by studying alkenes and ketones with saturation concentrations ($C^*$) ranging from $10^4$ to $10^7$ µg m$^{-3}$ in perfluoroalkoxy alkanes (PFA) Teflon tubing. The delay times due to gas–tubing wall partitioning increased strongly as $C^*$ decreased, and were well described by a model analogous to retention times in a gas chromatography column. The model source code was made publicly available to facilitate inlet design. In those studies, the environmental chamber walls and Teflon inlet walls were treated as an equivalent absorbing organic mass ($C_w$, µg m$^{-3}$). The results for tubing and chambers were consistent (including the determination of $C_w$ after scaling by the surface volume ratio, S/V), but tubing is a far superior system for determining parameters for gas/surface interactions, due to the much higher S/V and the removal of the effect of variations in buoyancy and turbulent transport which complicate the interpretation of chamber results (Krechmer et al., 2016).

Deming et al. (2019) have extended the study of the same VOCs and IVOCs as in Pagonis et al. (2017) to many other commonly-used tubing materials including different types of Teflon and other polymer tubing, as well as uncoated and coated stainless steel, glass, aluminium and others. All polymeric tubing showed responses consistent with *absorptive* partitioning, with PFA Teflon performing best. Uncoated and coated metal and glass tubing, on the other hand, resulted in longer delays and had responses consistent with *adsorptive* partitioning. Delays due to adsorptive interaction can be reduced by using relative humidities above 20%. Due to the finite number of surface sites for adsorptive partitioning,





memory effects and concentration-dependent responses were observed, which makes the modeling of analytical systems much more challenging.

The above studies have not included species in the lower part of the IVOC range ($10^3 < C^* < 10^4$ µg m$^{-3}$) or semi-volatile species (SVOCs, $C^* < 10^3$ µg m$^{-3}$). Application of the Pagonis et al. (2017) model suggests that the delays will increase continuously as $C^*$ decreases. Since semi-volatile gases are frequently in equilibrium with aerosols in different systems, delays in transport of gases can perturb the

gas/particle equilibria and lead to aerosol evaporation. In addition, species sampled in prior experiments and deposited in tubing or reactor walls may evaporate and re-condense onto particles, leading to complex multiphase memory effects. Measurements of potential aerosol mass using oxidation flow reactors and other systems often require inlets and can be severely perturbed by tubing delays as potential aerosol mass typically has a major contribution of lower volatility species (Li et al., 2013; Hunter et al., 2017; Ma et

al., 2017; Palm et al., 2018). Therefore, the partitioning of semivolatile gases in tubing affects both gas and aerosol quantification and needs a better understanding.

SVOCs can be monitored in-situ using soft ionization mass spectrometry. Delays in instrument response have been reported for proton-transfer-reaction mass spectrometers (PTR-MS) for VOCs and for IVOCs with $C^*$ as low as $3\times 10^4$ µg m$^{-3}$ (Pagonis et al., 2017; Krechmer et al., 2018). While chemical

ionization mass spectrometers (CIMS) measure even lower-volatility compounds and are commonly used (Jokinen et al., 2012; Lee et al., 2014), to our knowledge CIMS instrument delays have not been systematically and quantitatively characterized.

Besides S/IVOCs, some small molecules, such as HNO$_3$ and NH$_3$, readily absorb or adsorb to surfaces, and for these species particular care needs to be taken to ensure high passing efficiency and fast

time response (Huey et al., 2004; Nowak et al., 2007). PFA and FEP inlets were recommended for use as sample lines and were sometimes combined with special flow design and/or heating to minimize delay (Neuman et al., 1999; Nowak et al., 2007). HNO$_3$ can desorb from the CIMS ion-molecule reactor (IMR) or flow tube surfaces. The addition of NH$_3$ or coating treatment with sodium bicarbonate or other species can suppress the HNO$_3$ background by providing a large surface sink (Huey et al., 1998; Roscioli et al.,

2016). In summary, CIMS measurements of small trace gases can achieve fast time response when carefully designed; e.g., NH$_3$ with response time < 5 s, and HNO$_3$ and many other molecules including





peroxyacyl nitrates (PANs), molecular chlorine ($Cl_2$), nitryl chloride ($ClNO_2$) with response time < 1 s (Huey et al., 1998; Slusher et al., 2004; Nowak et al., 2007; Osthoff et al., 2008). These systems have been optimized one-at-a-time, however, and to our knowledge no systematic understanding of the

controlling parameters for tubing and instrument response exists for these molecules that could inform future experimental and instrumental designs.

This study extends the systematic and quantitative characterization of tubing and instrument delay times to SVOCs and to lower volatility IVOCs ($10^0$ to $10^4$ µg m$^{-3}$) using CIMS coupled with an I$^-$ source. These semivolatile organic compounds (SVOCs) were rapidly generated through photochemistry in a

Teflon chamber. Following on the results of Deming et al. (2019), the three best polymer materials, PFA, FEP, and conductive PFA; and the best metal, stainless steel, were selected as the testing inlets for the SVOCs. Two other materials were also included, Silonite as an example of passivated stainless steel, and Nafion since it is frequently used in dryers when sampling aerosols (and thus very relevant to SVOC sampling). The delays due to surfaces of CIMS instruments (mainly IMR surfaces) and tubing were

separately characterized. The Pagonis et al. (2017) model was employed to predict the delays and infer equivalent absorbing mass of these materials. In addition, we expand the range of analytes tested to include small polar organic and inorganic molecules. These results will enable improved and faster quantification of semivolatile species by informing inlet material selection and instrumental design, and also provide a useful method to quantitatively characterize any gas/surface interactions.

## 2 Experimental

### 2.1 S/IVOC experiments

Experiments for lower volatility species were performed in the CU Environmental Chamber Facility, using either 20 m$^3$ or 8 m$^3$ chambers under dry or RH=47% conditions. The semivolatile compounds were produced in the chambers using the experimental protocol described elsewhere

(Krechmer et al., 2016; Krechmer et al., 2017). Briefly, a series of 1-alkanols ($C_6$, $C_8$, $C_9$, $C_{10}$, and $C_{12}$) were injected into the chamber together with methyl nitrite and NO, and the UV blacklights were turned on for 10 s to produce rapid bursts of pptv levels of lower volatility products, including hydroxynitrates



(HNs), dihydroxynitrates (DHNs), and dihydroxycarbonyls (DHCs) (Table S1). The values of $C^*$ of these compounds were estimated using SIMPOL (Pankow and Asher, 2008). These compounds were sampled

through tubing of 0.62 m FEP (0.476 cm I.D.), 1.8 m PFA (0.476 cm I.D.), 1.0 m stainless steel (0.476 cm I.D.), 1.5 m conductive PFA (0.476 cm I.D.), 1.8 m Silonite (0.533 cm I.D.), or 0.60 m Nafion (0.178 cm I.D.), at flow rates of 4.0–4.2 L min$^{-1}$ for the first three materials and 2.6 L min$^{-1}$ for the latter three. In the laminar regime, higher flow rates tend to shorten response times (Pagonis et al., 2017), thus ~4 L min$^{-1}$ (Reynolds number ~1000 in 0.476 cm I.D. tubing) was preferred and used when instrument

configurations allowed. The differences in tubing length can be normalized for as described in Sect. 2.3. Three Aerodyne iodide-adduct time-of-flight CIMS (I-CIMS) instruments, two from CU-Boulder and one from the University of Washington (UW) were used to measure the semivolatile species (Lee et al., 2014; Krechmer et al., 2016). The two CU-Boulder instruments have slightly different IMR dimensions but were operated at the same IMR pressure (100 mbar) and flows, which consisted of an inlet flow and an

ion source flow each of 2 L min$^{-1}$. The UW instrument was equipped with a custom-designed IMR, which used different flow patterns and also used Teflon tubing to line the internal metal surfaces to reduce IMR response time to compounds such as $HNO_3$ and oxidized organics. A more detailed description of this design will be provided in a separate publication.

The instrument and tubing response times were measured by depassivation and/or passivation

procedures (Pagonis et al., 2017; Deming et al., 2019). For depassivation, after the instrument and tubing have equilibrated with the SVOC-filled chamber air (under constant sampling flow), the instrument alone or the instrument plus tubing was quickly switched to sample clean air. To minimize the effect of humidity change, the humidities of the SVOC-containing air and the clean air were either identical (two humidified chambers with RH difference < 1%) or similar (dry chamber air versus clean chamber enclosure air with

RH < 10%). Passivation tests were also performed for Nafion and metals (stainless steel and Silonite) to test transmission efficiency. Since uncoated metals were observed to respond faster when RH > 20% (Deming et al., 2019), a stainless steel inlet under RH=47% was tested. In this test, clean, humid (RH=47%) air was first flowed through the steel tubing until equilibration was reached before analyte sampling. Then we connected the RH-equilibrated steel line between the SVOC chamber and the

instrument. The sampling continued until a steady-state was reached. Tubing and instrument delays for



these SVOCs were quantified using depassivation and passivation measurements as described below in Sect. 2.3.

## 2.2 Experiments with small organic and inorganic molecules

The NOAA thermal dissociation chemical ionization mass spectrometer (TD-CIMS) (Osthoff et
al., 2008; Zheng et al., 2011) and the NOAA negative ion proton-transfer CIMS (NI-PT-CIMS) (Veres et al., 2008) were used for measuring delays of small polar organic and inorganic molecules. The tubing plus instrument response times were measured for a variety of small inorganic and organic compounds by introducing a step-function change in the analyte concentration produced by a calibration source. The inlets used for the small polar compounds differed according to each NOAA CIMS. The TD-CIMS used
0.63 cm O.D. (0.476 cm I.D.) PFA tubing of approximately 1 m length and a flow rate of 2.7 L min$^{-1}$ (Osthoff et al., 2008; Warneke et al., 2016). The NI-PT-CIMS used a 0.315 cm O.D. (0.159 cm I.D.) PFA tubing of 2 m length at a flow rate of 0.59 L min$^{-1}$ (Roberts et al., 2010). Despite these differences, the surface areas of the two inlets were within 40% of one another and the inlet residence times were similar, 0.377 s and 0.389 s for the TD-CIMS and the NI-PT-CIMS inlet, respectively. The sample streams were
humidified in the range of 20–50% RH. No significant dependences of compound equilibration times were observed over this RH range. However, equilibration times were faster under dry inlet gas conditions for these compounds, a feature that has been reported before for strong acids (e.g., Neuman et al. (1999)). The conditions of the IMRs were roughly the same, as the ion sources used 2 standard L min$^{-1}$ of $N_2$ through the Po-210 ionizer, with a small addition of reagent gas. The IMRs were pressure controlled at
33 to 40 mbar. The measured responses to these changes were fitted to single exponential curves to determine the total tubing and instrument response timescale.

## 2.3 Delay quantification

The depassivation procedure described above was used to quantify instrument and tubing delays, during which the decrease in signals can be fitted as double or triple exponential decays. Instrument delays
were defined as the time it takes each species to reach 10% of the signal measured at the beginning of depassivation. Since tubing depassivation must be measured together with the instrument, we defined the



tubing delay times as the difference between the time to reach 10% signals that of instrument plus tubing and that of the instrument only. Both definitions are consistent with previous studies that quantified delay times for higher volatility compounds (Pagonis et al., 2017; Krechmer et al., 2018; Deming et al., 2019).

Since Pagonis et al. (2017) found that tubing delay time increases linearly with tubing length, we normalized each tubing delay time by the corresponding length used to 1 m. On the other hand, as shown in Table 1, the sampling flow rates also varied, which are expected to affect tubing delay time nonlinearly and thus cannot be simply normalized (Pagonis et al., 2017). However, a model that treats the gas-wall interaction in tubing using gas chromatography principles (Pagonis et al., 2017) was used to derive the

compound and tubing material specific $C_w$ and then predict tubing delay times under a universal flow rate.

## 3 Results and Discussion

### 3.1 Instrument response

The instrument response time as a function of compound $C^*$ for a number of instruments were

quantified in this work. Figure 1 shows examples of the signal decrease for three compounds during depassivation with or without FEP tubing attached, together with exponential fittings. Double or triple exponential decays were also observed for other CIMS used in this work for the same set of semivolatile compounds, consistent with prior results for a Vocus proton-transfer time-of-flight mass spectrometer (Vocus PTR-TOF) for more volatile ketones ($C^*$ $10^4$-$10^7$ µg m$^{-3}$) (Krechmer et al., 2018). The initial fast

decay is likely due to fast clearing of the instrument and tubing. The slower parts of the time response depend on re-partitioning of compounds from the instrument and tubing walls into air.




**Table 1.** Instruments and tubing materials tested in this study.

| Instrument | IMR H$_2$O (molec cm$^{-3}$) | Tubing | Gas flow rate (L min$^{-1}$) | Inner diameter (cm) | Tubing RH (%) | Tubing length (m) | Tubing supplier |
|---|---|---|---|---|---|---|---|
| CU (Jimenez) I-CIMS | ~4×10$^{13}$ | FEP | 4.0 | 0.476 | 0 | 0.62 | Saint-Gobain |
| CU (Ziemann) I-CIMS | 2×10$^{16}$ | PFA | 4.2 | 0.476 | 47 | 1.8 | McMaster-Carr |
| | | Stainless steel | 4.2 | 0.476 | 47 | 1.0 | McMaster-Carr |
| UW I-CIMS | ~1-2×10$^{16}$ | Conductive PFA | 2.6 | 0.476 | 0 | 1.5 | Fluorostore |
| | | Silonite | 2.6 | 0.533 | 0 | 1.8 | Entech Instruments Inc. |
| | | Nafion | 2.6 | 0.178 | 0 | 0.60 | Perma Pure |
| NOAA TD-CIMS | wet[a] | PFA | 2.7 | 0.476 | 20 - 50 | ~1 | Saint-Gobain |
| NOAA NI-PT-CIMS | ~2×10$^{15}$ | PFA | 0.59 | 0.159 | 20 - 50 | 2 | Saint-Gobain |

[a] Water added to IMR by bubbling 10 standard mL min$^{-1}$ of N$_2$ through water.


The measured I-CIMS delay times (times to decrease to 10% of initial signals) as a function of analyte $C^*$ are shown in Figure 2. Also plotted are delay times for a series of ketones measured by a quadrupole proton-transfer mass spectrometer (q-PTRMS) (de Gouw and Warneke, 2007), the q-PTRMS with a simplified inlet system, and a Vocus PTR-TOF (Krechmer et al., 2018). Simplifying the inlet on

the q-PTRMS decreased the delay in the response by a factor of five for the least volatile ketone (2-tetradecanone). The Vocus PTR-TOF (Krechmer et al., 2018) was an order of magnitude better than even the improved q-PTRMS across the whole $C^*$ range due to substantial reduction of the instrument surfaces exposed to the inlet gases. The time response of the Jimenez I-CIMS appears to be consistent with that of the Vocus PTR-TOF, extending to the lower $C^*$ measurable with this instrument.



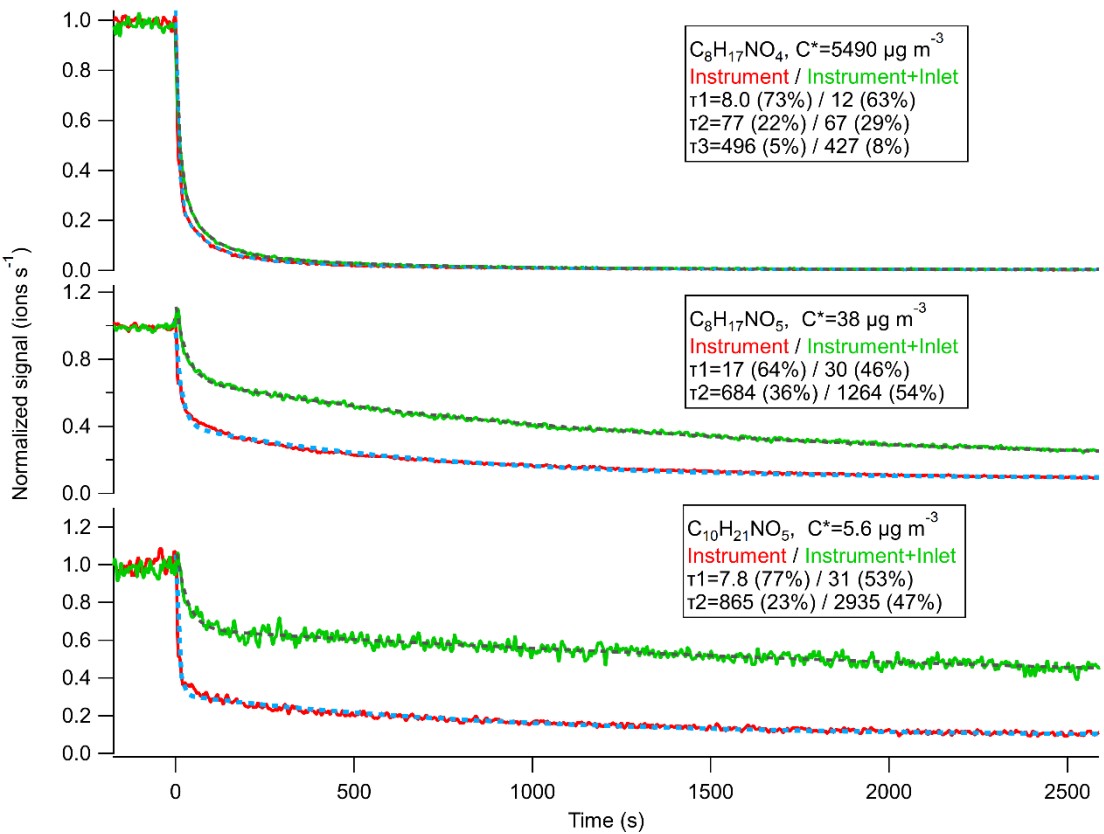

**Figure 1.** *Example instrument-only (red, solid lines) and instrument+FEP tubing (green, solid lines) depassivation responses for 3 different hydroxynitrates observed by the Jimenez I-CIMS. In all cases multi-exponential decays were observed (dashed line fits), with a substantial initial fast decay, followed by a much slower decrease of the signal. The percentages in parentheses are fractions of original signals associated with each timescale.*



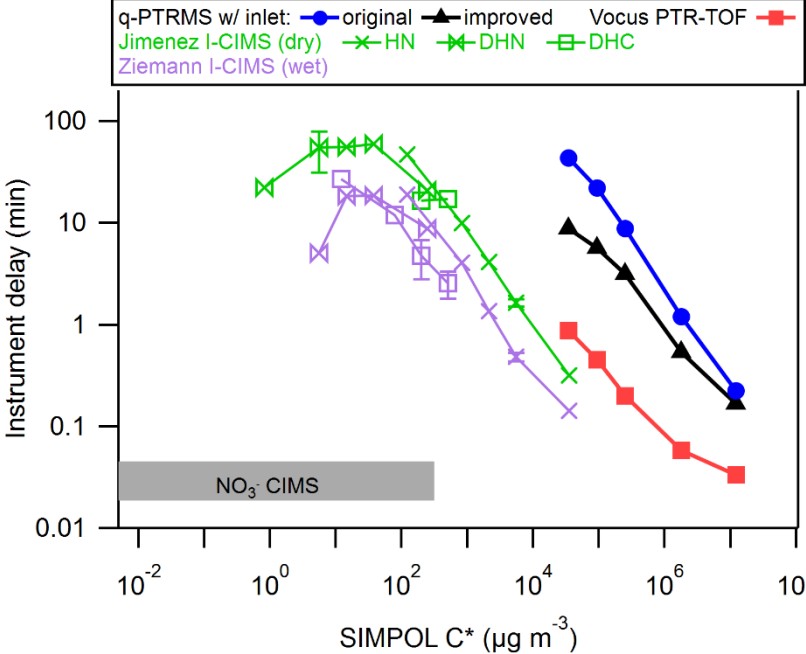

**Figure 2**. *Measured instrument delay times as a function of SIMPOL C\* for five instruments. The q-PTRMS data are from Pagonis et al. (2017) and the Vocus PTR-TOF data are from Krechmer et al. (2018). Example error bars are shown, calculated as standard deviations of two or three repeated measurements. Also shown are approximate delays (shaded area) observed with the $NO_3$-CIMS in Krechmer et al. (2016).*


According to the instruments' reactor dimensions and operating conditions, the residence time in the CIMS reactor is between 1 to 2 orders of magnitude faster than its radial diffusion timescale (similar to the Vocus PTR-TOF). From Figure 2 it is clear that as $C^*$ decreases, the delay times increase until $C^*$ reaches ~100 μg m$^{-3}$, from which point the delay times start decreasing. We interpret this trend when $C^*$

< ~100 μg m$^{-3}$ as being due to irreversible losses of species to the instrument surfaces (at least within timescales relevant to these experiments). The Ziemann I-CIMS showed ~3 times faster response than the Jimenez I-CIMS. Since the dimensions of the two IMR regions of the two instruments are very similar, we attribute the improved response to the humidity added to the stainless steel IMR of the Ziemann I-CIMS. This is supported by the finding by Deming et al. (2019) for stainless steel tubing, that an increase



of relative humidity from 0% to 20% could decrease the delay time by a factor of ~10. Also shown in

Figure 2 are earlier data acquired for the Jimenez CIMS coupled to a $NO_3^-$ source (Krechmer et al., 2016).

The $NO_3^-$ source has a much faster response time of less than two seconds, which is due to the combination

of using a sheath flow in its ion-molecule reaction region, as well as to only measuring species in the $C^*$

range where species are likely irreversibly lost to walls. We suggest comparing future instruments and

instrument modifications in a similar way to enable objective comparisons in their responses and we have

made this graph available as an Igor file in the supplementary information to facilitate future comparisons.

In general, instrumental response times can be improved by minimizing analyte surface contact.

**3.2 Teflon tubing delays for semivolatile compounds**

The measured tubing delays of HNs, DHCs, and DHNs through Teflon inlets, PFA, FEP, and

conductive PFA, are shown in Figure 3. A few low-volatility compounds were excluded, compared to

Figure 2, since their delay times cannot be accurately quantified because the tubing partitioning is too

slow. Conductive PFA is made from PFA with added black carbon to make the tubing conductive and

thus protect from static electricity discharges in settings where flammable gases are present. Teflon tubing

typically experiences dramatic losses of all charged particles, and for that reason metal tubing (copper

and stainless steel typically) is used for aerosol sampling. A conductive silicone tubing is commercially

available (TSI. St. Paul, MN), but it has significant artefacts due to condensation of plasticizers onto

particles, and thus it is not recommended for sampling ahead of aerosol chemistry instruments (Timko et

al., 2009; Yu et al., 2009). As shown by Deming et al. (2019), conductive PFA can sample aerosols

without losses, provided that it is protected from rubbing that builds up static electricity (e.g. by adding

an aluminium foil cover). As shown in Figure 3, at similar flow rates (~4 L min$^{-1}$), PFA performs better

than FEP in terms of delay times for S/IVOCs, consistent with the results for higher volatility species in

Deming et al. (2019). Conductive PFA was tested at a slower flow rate of 2.6 L min$^{-1}$, so cannot be

compared directly to the other two materials due to nonlinear dependence on flow rates. Using the gas

chromatography model by Pagonis et al. (2017), we fit the modeled delay time to the measurements by



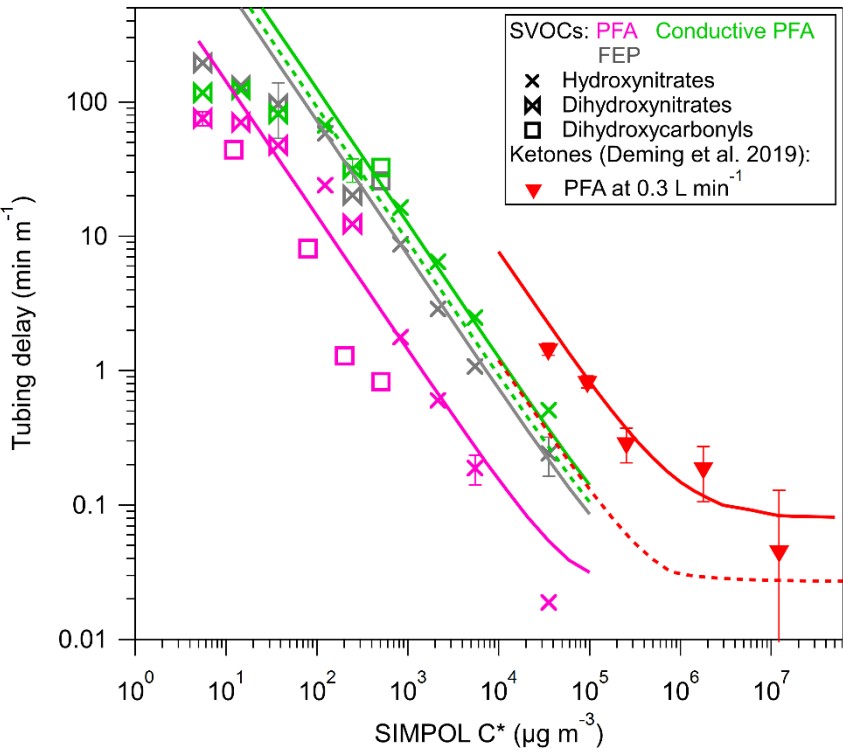

**Figure 3.** *Measured tubing delays normalized by tubing length as a function of the saturation concentration of the gas compounds. Also shown as a comparison is the PFA result by Deming et al. (2019). Example error bars are shown, as propagated from exponential fits. Solid lines are chromatographic model results using the best estimated $C_w$ (for $C^* = 10^2$–$3\times10^4$ µg m$^{-3}$) for different tubing at the same flow rates (Table 1). Dashed lines are model results for conductive PFA (this study) and PFA (Deming et al. (2019), using the $C_w$ determined in that work) at 4 L min$^{-1}$.*

optimizing $C_w$ (Table 2). With the estimated $C_w$, the model then simulated delay times for each material under the same 4 L min$^{-1}$ flow rate. The model run results in Figures 3 and S2 show that the delay time of conductive PFA was slightly shortened from 2.6 L min$^{-1}$ to 4 L min$^{-1}$ and that conductive PFA has similar delay as FEP within the uncertainties. In addition, an inspection of the I-CIMS (and Vocus PTRMS (Deming et al., 2019)) mass spectra obtained using conductive PFA indicates no obvious gas emissions from the tubing compared to PFA, PTFE, and FEP. Given that conductive PFA works well for



**Table 2.** Fitted values of $C_w$ (µg m$^{-3}$) for S/IVOCs of three $C^*$ ranges (µg m$^{-3}$) into Teflon tubing materials. $C_w$ needs to be scaled by the S/V ratio of the system of interest when applying the recommended values (Pagonis et al. (2017)).

| Tubing | $C_w$ | $C_w$ | $C_w$ | Deming et al. $C_w$ | Internal S/V ratio (cm$^{-1}$) |
|---|---|---|---|---|---|
| $C^*$ range | < 5 | 5 - 10$^2$ | 10$^2$ - 10$^4$ | 10$^4$ – 10$^7$ | |
| PFA | $2.7 \times 10^4$ | $1.16 \times 10^4 (C^*)^{0.56}$ [b] | $9.0 \times 10^4$ [a] | $8.0 \times 10^5$ | 8.40 |
| FEP | $1.2 \times 10^5$ | $8.46 \times 10^4 (C^*)^{0.38}$ | $4.9 \times 10^5$ | $2.0 \times 10^6$ | 8.40 |
| Conductive PFA | $4.4 \times 10^4$ | $1.02 \times 10^4 (C^*)^{0.85}$ | $6.2 \times 10^5$ | $1.3 \times 10^6$ | 8.40 |

[a] Dihydroxycarbonyl (DHC) compounds excluded due to potential humidity effect. This $C_w$ value is applicable for $C^*$ as low as 40 µg m$^{-3}$.

[b] The equation works for $C^*$ 5 - 40 µg m$^{-3}$.

aerosol sampling (Deming et al., 2019), it appears to be an excellent choice when concurrent semivolatile gas and aerosol sampling is needed.

Besides the above polymers, Nafion polymer material was also tested, since it is commonly used
in sampling systems. Nafion is similar to Teflon but it is modified by adding sulfonic acid groups that facilitate transfer of water and cations across the tube walls. Nafion is commonly used to transfer water vapor by permeation from a humid gas stream to a drier purge gas stream. During a passivation test for the S/IVOC compounds, the Nafion inlet initially transmitted only 10-30% of the concentrations of these hydroxyl group-containing compounds, without further increase with time (Figure S1). Once detached
from the SVOC-filled air, depassivation was immediate. The sulfonic acid groups likely change the absorptive properties of Nafion compared to other Teflon by adding hydrophilic regions in addition to the hydrophobic Teflon backbone. Thus, the interaction between the Nafion surface and hydroxyl groups of the analytes appears to result in large irreversible losses. Therefore, we do not recommend the use of Nafion in sampling inlets for polar S/IVOCs. Aerosols in equilibrium with S/IVOCs will be perturbed by



the removal of the gases by the Nafion, so we recommend installing Nafion dryers or humidifiers at the last possible location before an instrument and to minimize residence time in both the dryer and between the dryer and instrument whenever possible.

The estimated $C_w$ values that allow the model to reproduce the observed delay trends for the three Teflon tubing materials is shown in Table 2. As shown in Figure 3, the delay time of species with $C^* <$
$10^2$ µg m$^{-3}$ generally increases less as $C^*$ decreases compared to more volatile species. Thus, we fitted $C_w$ in $C^* > 10^2$ µg m$^{-3}$ and $C^* < 10^2$ µg m$^{-3}$ ranges, respectively. Also listed in Table 2 are $C_w$ values estimated for higher volatility ketones (Deming et al., 2019). $C_w$ values fitted in this work are consistently smaller than those fitted for more volatile ketones (Table 2 and Figure S2). This difference may be due to several effects: (1) differences in activity coefficients for absorption in Teflon  (Krechmer et al., 2016; Huang et
al., 2018), i.e., it is likely that the compounds of lower $C^*$ tested in this study have higher activity coefficients in the non-polar Teflon "solvent" than the monofunctional ketones tested by Deming et al. (2019) (Figure S3); (2) possible uncertainties in vapor pressures calculated using SIMPOL for multifunctional compounds; and (3) differences in vapor pressures across positional isomers that CIMS and SIMPOL cannot distinguish. The partitioning of the same S/IVOC species to an FEP chamber bag
was previously investigated in Krechmer et al. (2016), and our $C_w$ for FEP tubing is of the same order but not identical (Figure S3). Differences could be due to differences in the FEP materials and mixing effects in the tubing and chamber. Additionally, humidity (RH=47% in our case) appears to have decreased the activity coefficients of DHC compounds in PFA tubing, as indicated by their shorter delay times compared to other organic nitrates of similar vapor pressures. Another possibility is that humidity affected
the extent to which DHCs cyclized to hydroxy cyclic hemiacetals on the walls, which then affected DHC partitioning (Lim and Ziemann, 2009). DHC compounds reacted to humidity similarly in FEP tubing, although the data is not shown here because the humidity in the chamber and clean air was not controlled to be exactly the same in that experiment. The possible humidity effect was not seen for organic hydroxynitrates, nor in previous work for ketones (Deming et al., 2019).



## 320    3.3 Stainless steel and Silonite tubing delays for semivolatile compounds

Two metal tubing materials, stainless steel and Silonite, were tested. Stainless steel was chosen because it exhibited the fastest response for ketones among a variety of metals and glass (Deming et al., 2019), and also because it is a typical tubing material when sampling aerosols containing semivolatile primary and secondary organics. Silonite (Entech Instruments Inc.) is a ceramic coating treatment applied
onto metals to provide a smooth, inert surface that reduces the potential for chemical adsorption/absorption. Deming et al. (2019) observed that the passivation of metal tubing with a mixture of linear ketones resulted in complex behaviour in which the less volatile species competitively displaced the more volatile species from the surface sites, in some cases enriching the gas-phase concentrations of the displaced compounds several times above the chamber concentration. In our experiments, the stainless
steel and Silonite tubings were attached to the chamber containing ~1.4 ppm of alkanol precursors and the SVOCs formed from the reaction of only ~3.6% of the total alkanols, as measured by the CIMS. Thus, the passivation behavior observed in this experiment is a result of the flow of the S/IVOCs over tubes that had presumably already equilibrated with the alkanols (which were not measured here). In other words, a significant fraction of surface sites might have been occupied by the high-concentration alkanols, so the
targeted SVOCs equilibrated faster with the remaining sites. In the atmosphere or other experiments with less concentrated compounds, the displacement behavior might still occur. In general (Figure 4), under our conditions stainless steel and Silonite exhibited similar passivation and depassivation behaviors for the compounds studied. For stainless steel tubing (RH=47%) and for more volatile species ($10^2 < C^* < 10^5$ $\mu g\ m^{-3}$), the passivation time needed increased with decreasing volatility (Figure 4a). A slight enrichment
effect was observed for the most volatile species ($C_6$ HN), similar to results from Deming et al. (2019) and presumably due to competitive displacement by the less volatile species. An irreversible loss was observed for $C^* < 10^2$ $\mu g\ m^{-3}$ (including $C_{12}$ HN and $C_8$ and higher number carbon DHNs) after ~5 min of passivation, corresponding to a transmission efficiency of only ~50%. Similarly, irreversible loss also appeared to occur for this lower volatility range within the stainless steel IMR of the CIMS instrument
(Figure 2). The volatility dependence seemed to be even stronger for the dry passivation of Silonite (Figure 4c), since the passivation time reached almost 1 h for $C^* \sim 10^3$ $\mu g\ m^{-3}$ species and significant losses occurred for $C^* < 10^3$ $\mu g\ m^{-3}$ species. The slower sampling flow rate through Silonite (2.6 L min$^{-1}$)



compared to stainless steel (4.2 L min$^{-1}$) may partly explain its poorer performance. The depassivation of

the two metal tubings resemble that of Teflon tubings (Figure 4b,d). Note that the RH was the same before

and after depassivation, so that we did not observe the major effect from competitive water displacement

**Figure 4.** *Examples of passivation and depassivation of stainless steel ((a) and (b)) for $C_6$-$C_{10}$ HNs and DHNs and Silonite ((c) and (d)) for $C^* > 10^2$ µg m$^{-3}$ HNs and DHNs, where curves were binomially*

*smoothed across 10 points for visual clarity. The stainless steel data were measured by the CU (Ziemann) CIMS. The Silonite data were measured with the UW CIMS, and the gaps were due to periodic*





*background sampling. Note that the starting signals for (c) are not zero, due to compounds that remained*

*in the instrument.*

that Deming et al. (2019) reported for metals when dry, ketone-filled air was replaced with clean, humid room air. This may be due to the higher polarity for the species in our study, which may compete favourably with water for surface sites.

      The measured delay times for compounds with transmission efficiency of 100% are shown in Figure 5. The two metal tubing materials resulted in longer delays when compared to other Teflon

materials. In summary, metal tubing transmits IVOCs but with longer delays than Teflon tubing. In addition, metal tubing appears less suitable than Teflon for SVOC measurements due to the relatively long delays and potentially irreversible losses leading to low transmission. This may be a problem when sampling particles in equilibrium with SVOCs, as the depletion of the gas-phase due to the delays and losses may lead to evaporation of those compounds from the particles (given similar timescales), and thus

a negative bias on the particle measurements.



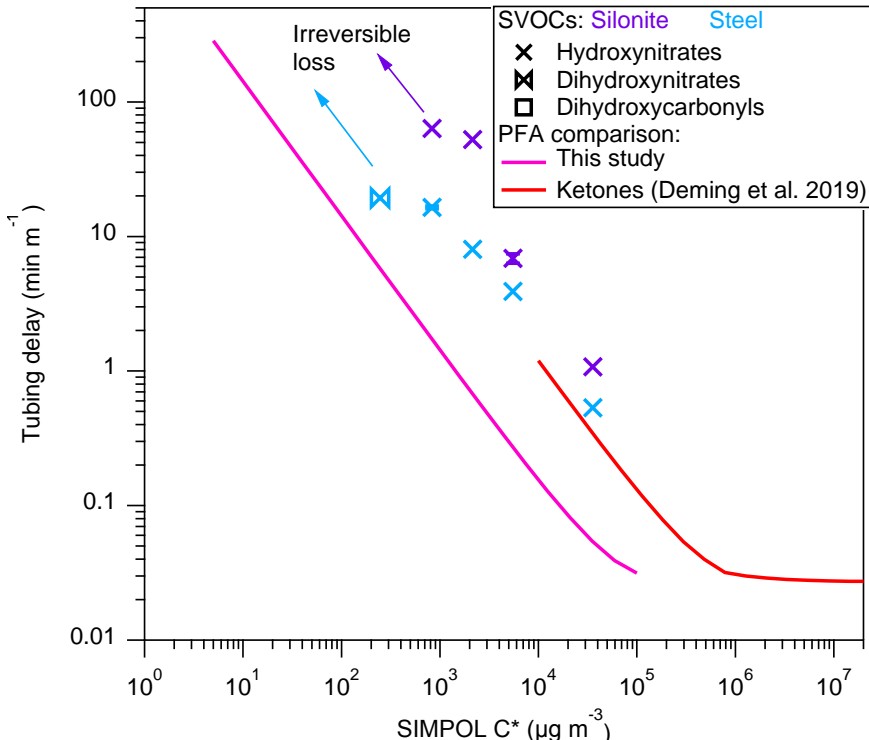

**Figure 5.** *Stainless steel and Silonite depassivation delay times measured for a series of HNs, DHNs and DHCs after equilibration with a chamber filled with alkanols and S/IVOCs, and compared with PFA modeled delays for 4 L min$^{-1}$ flows. Only compounds with 100% transmission efficiency through stainless steel and Silonite compared to Teflon tubing are shown. Example error bars (smaller than marker) were propagated from exponential fits.*

### 3.4 Tubing delays for small polar compounds in Teflon tubing and instruments

The timescales for passivation of the inlet tubing and the instruments by small organic and inorganic molecules are shown in Figure 6. $C^*$ does not show a relationship with delay time (defined as the time constant of a single-exponential fit in this case), in contrast to the results for larger organic molecules (Figure 6a). Instead, there is a clear relationship between the response timescale observed and the Henry's Law coefficient of each compound (Figure 6b). Several of the small polar compounds are

weak acids and so have solubilities that depend on pH according to the following relationship:



$$H_{eff=}H^*(1 + \frac{K_a}{[H^+]})$$  (1)

where $H^*$ is the intrinsic Henry's coefficient and $K_a$ is the acid dissociation constant for the weak acid. Figure 6b was plotted using $H^*$ for HNCO and $HNO_2$ (Sander, 2015), as this was most appropriate for the small quantities of absorbed water that created this surface effect.

The delay times of these small polar species through tubing are due to absorption into small amounts of liquid water present on/in the Teflon or IMR walls under these sampling conditions (~20-50% RH). The amount of liquid water can be estimated from the observed delays and the chromatography model of Pagonis et al. (2017), and is equivalent to 1.5–150 µL. Since the instrument and tubing were not tested separately for these species, this value represents the combined volume of water for the tubing

surface and the surfaces of the instrument (including the IMR region). Assuming that all the water is located on the surface of the Teflon tubing, we estimate the range of water content in these experiments to be 0.1–10 mL of water per square meter of tubing, equivalent to a film thickness of ~0.1-10 µm. As this is a substantial thickness, especially at the upper limit, it is likely that the IMR region plays a role in the observed delays. Details of how the model of Pagonis et al. (2017) was adapted to estimate liquid

water in tubing are presented in the Supplementary Information. In contrast to the organic compounds discussed above, small polar compounds have longer delay times under humidified conditions than under dry conditions even for materials such as PFA Teflon. In addition, this effect is expected to be enhanced for steel and glass due to their hygroscopic nature. Therefore when optimizing measurement response time for such small polar molecules, ambient humidity or water added on purpose (e.g., for enhancing

CIMS sensitivities towards peroxyacyl nitrates, etc. (Slusher et al., 2004)) needs to be considered in addition to tubing and instrumental configurations.





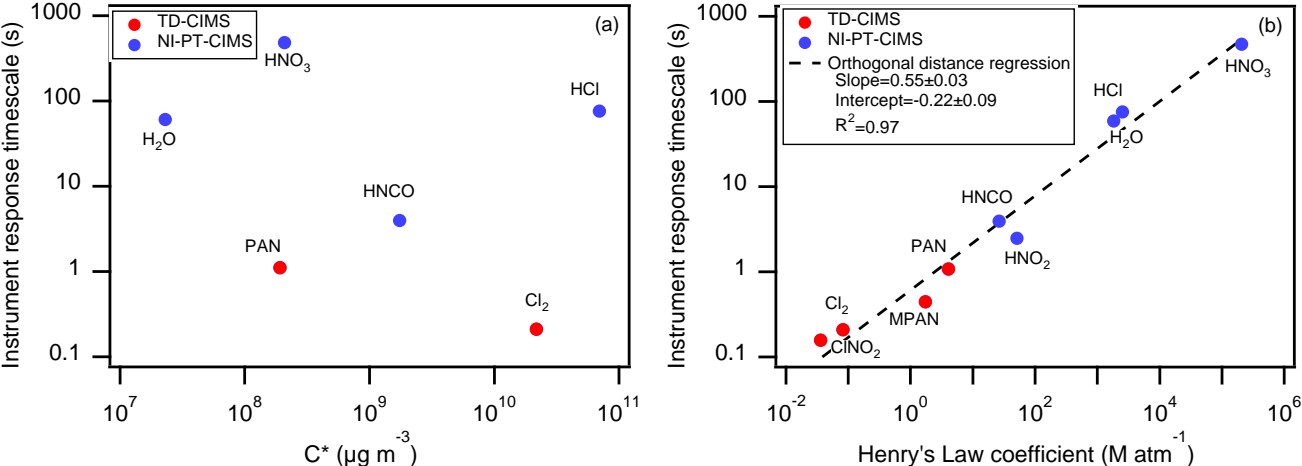

*Figure 6. Instrument response timescales as a function of (a) saturation concentration and (b) Henry's Law coefficient for several inorganic compounds measured using TD-CIMS and NI-PT-CIMS. The dashed line is an orthogonal distance regression of the logarithm of the response timescale against the logarithm of Henry's Law coefficient. Peroxyacetyl nitrate and peroxymethacryloyl nitrate are abbreviated as PAN and MPAN.*

## 4 Conclusions

The instrument and tubing delays of S/IVOCs with saturation concentration between $10^0$ to $10^4$ µg m$^{-3}$ were characterized, which are useful for the design of improved inlets and instruments. This technique is also useful for improved characterization of surface/gas interactions for other applications. In the standard IMR regions of I-CIMS that are commonly used, the instrument response time increases with decreasing volatility until an apparent irreversible loss emerges when $C^* < 10^2$ µg m$^{-3}$. Humidifying the metal IMR region was found to help shorten I-CIMS response time, presumably due to water molecules occupying some adsorption sites. As CIMS and other soft ionization techniques have been widely used for monitoring semivolatile, multifunctional organic compounds, delay characterization with methods similar to this study is recommended. Future improvements minimizing surface contact are needed for fast measurements and accurate quantification of a wide range of S/IVOCs. The NO$_3$-CIMS is an exception, since because of its sheath flow design and measurement limitations to low $C^*$ species



that are likely irreversibly lost to walls it can measure SVOCs and LVOCs with minimal delays. However, the NO$_3$-CIMS can only detect a narrow spectrum of highly oxidized compounds (Hyttinen et al., 2018). If possible, adopting a similar flow design in front of other types of CIMS could potentially improve overall time response.

Among the tubing materials tested, the tubing delay time for the S/IVOCs analyzed increased in the order of PFA, FEP, conductive PFA, wet stainless steel, and Silonite. Irreversible loss was observed for compounds with $C^* < 10^2$ µg m$^{-3}$ for stainless steel, $C^* < 10^3$ µg m$^{-3}$ for Silonite, and the whole $C^*$ range studied here for Nafion. Thus, PFA Teflon tubing is recommended over the other tested materials for use in atmospheric measurement sample lines when particle transmission is not a consideration (or in

specific laboratory experiments where the particles are not appreciably charged). Conductive PFA appears to be the best compromise for simultaneous gas and particle sampling, as it can be easily protected from build-up of static electricity to prevent particle losses, and its delay time is the smallest of the tubing materials that transmit particles. We modeled the delays caused by Teflon tubing using a simple chromatography model (Pagonis et al., 2017) and estimated the effective absorbing mass concentration

($C_w$) of each material. The $C_w$ values can be potentially used for estimating delay times under different flow rates and tubing dimensions; however, the applicability of the $C_w$ values reported here to compounds with very different functionality may need to be evaluated and modified to account for activity coefficient differences.

In addition, we have only studied one tubing sample for each material, and it is possible that some

differences occur between tubing materials from different manufacturers. Surface conditioning due to sampling history may also affect the observed delays for metal materials, but this is unlikely for Teflon (Matsunaga and Ziemann, 2010).

We have not explored the use of higher tubing temperatures to improve transmission, which is commonly applied for semivolatile or reactive species (Mikoviny et al., 2010). Since ~15°C are needed

to increase $C^*$ by one order-of-magnitude (Epstein et al., 2010), however, very high temperatures may be required for fast transmission in the SVOC range (e.g. to increase $C^*$ from 10 to 10$^6$ µg m$^{-3}$ to try to ensure fast transmission, T ~ 100°C). Such high temperatures can lead to thermal decomposition of oxidized molecules (Stark et al., 2017) and potentially changes in the tubing properties. Heating also will

lead to evaporation of organic aerosol particles, typically ~1% of the mass per 1°C for ambient particles
(Huffman et al., 2009). The evaporated compounds will be at the lower end of the SVOC range, and thus
subject to irreversible losses and/or very long delays in the gas-phase compared to very efficient
transmission in the particle phase. Thus heating can help in certain cases, but it is unlikely to solve most
of the delay problems reported here, and can create other problems in the process.

A different phenomenology was observed for the sorption of small polar molecules, and appears
to be due to absorption into small quantities of water on/in the tubing and instrument walls since it
correlates with Henry's Law constants rather than $C^*$. Although this work and that of Deming et al. (2019)
did not observe any water solubility effect for the IVOCs and SVOCs studied, it could play a role for
smaller, more polar organic gases (e.g. formic acid). Simultaneous partitioning to adsorbed water and
polymer walls can be incorporated into tubing delay models in an straightforward manner, as done in past
studies for gas/particle partitioning (Yatavelli et al., 2014; Wania et al., 2015). Further experimental
investigation for compounds expected to have a range of fractional partitioning to both phases is needed
to confirm the applicability of that method.

The methods employed in this study are recommended for characterizing the response times and
transmission efficiencies of S/IVOCs and other compounds in different tubing materials and instruments
since both of these can affect the accuracy of quantification, especially when no direct calibration is
available. These results should help to inform inlet and instrument designs, guide the exploration of the
effects of other variables such as compound functionality and inlet temperature, and guide the evaluation
and design of future tubing materials and instrumentation.

**Acknowledgements**

This work was supported by the Department of Energy (BER/ASR) DE-SC0016559, the Sloan
Foundation program on the Chemistry of Indoor Environments (CIE, Grant 2018-10071), and the
NOAA's Health of the Atmosphere Program and NOAA's Climate Goal. We thank the ToF-CIMS user's
community for useful discussions.





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
