# Peer review of "Effects of Gas-Wall Interactions on Measurements of Semivolatile Compounds and Small Polar Molecules"

_Atmospheric Measurement Techniques, 2019_

## Referee Comment (RC1) · Anonymous Referee #2 · 29 Mar 2019

Liu et al. present a detailed characterization of the signal delay in detecting organic vapors with saturation vapor concentration ($C^*$) of $10^0 - 10^4 \mu$g m$^{-3}$ through different types of sampling tubes. Different types of CIMS have been used in this study to compare the effect of inlet design on signal delay in detection. RH effect is also probed. Adsorption or absorption of organic vapors by the tube wall under different situations are discussed. The characterization can be very helpful in designing an instrument's inlet for the detection of a fast-changing environment or quantification of gas-phase components. This manuscript is well-written and organized. I suggest for publication after considering the following aspects:

[Figure]

General comments about experimental suggestions:

Though detailed suggestions have been given in Section 4, I am concerning some other points that the authors have not covered. For example, given the much slower desorption time of low volatile species (Fig. 1 and 4b), are we supposed to use new sampling tubes for every experiment? In Fig. 1, the instrument-only signal decay of the compound $C_{10}H_{21}NO_5$ has not goes back to 0 in 2500 s. Does that mean the IMR has to be cleaned every time after detection of these species? Humidifying IMR shortens the response time based on this study, but will that affect the ionization efficiency of the vapor molecules? The other question is how to use the fact that low volatile species will level off after a while but with a low transmission efficiency, which could be an inverse problem for experiments without prior knowledge. I am sure the authors have the solution, but maybe a step-by-step process helps the readers a lot. The last is how confident the authors are with the relationship between $C_w$ and $C^*$. As the authors have shown the study of small polar molecules, how about the effect of functional groups of organic species? pH of the water film could play a role, how about the potential hydrolysis reactions?

Specific comments:

1. Page 7 Line 172: Usually in the exponential fitting, the decay rate is $k = \frac{1}{\tau} = \sum \frac{1}{\tau_i}$. Though 10% is defined in this study, I would use the same expression.

2. Page 7 Line 186: About double and triple exponential fitting: what is the $\tau$ value reported in this paper?

3. Figure 2: Only several points have error bars. Do all of them have error bars or do other data points simply have small error bars?

4. Figure 4: The red curve (C6HN) higher than 1 is explained by competitive replacement by less volatile compounds, but how to explain the decrease? Though compounds with $C^* < 100$ $\mu$g m$^{-3}$ have lower transmission efficiency, how to explain the

fast response time (the overlap with species of higher $C^*$ at the beginning)?

5. Figure 5: Looks like Dihydroxycarbonyls are not in the figure.

---

## Referee Comment (RC2) · Anonymous Referee #1 · 29 Mar 2019

This study investigates the time responses of semivolatile and intermediate-volatility organic compounds (S/IVOC) for different instrument inlet and tubing materials. The measured delay times could be explained by absorptive partitioning. The same model as developed earlier for VOCs could be applied for this data set by adjusting the material specific parameters. The results and the framework presented here is extremely useful for a proper design of instrument inlets and choice of tubing material to measure quantitatively low volatility multifunctional compounds. In a second part the authors also found that instrument response delay times for small polar molecules could be scaled with their Henry's Law coefficient. They partition to small amounts of water on the surfaces of the inlet or tubing. The manuscript is well written and data and results

are clearly presented. The manuscript can be published as is. I have only a few minor comments. Line 227: you mean: residence time is 1 to 2 orders of magnitude shorter than diffusion time scale Figure 5: DHC measurements are mentioned but not shown. Figure 6a: Could you indicate roughly the expected response time for these volatility classes; basically an extrapolation of Figure 2?

---

## Author Comment (AC1) · 17 May 2019

We thank the reviewers for their careful reading and their constructive comments on our manuscript. To guide the review process we have copied the reviewer comments in black text. Our responses are in regular blue font. We have responded to all the referee comments and made alterations to our paper (**in bold text**).

Reviewer #1
R1.0. This study investigates the time responses of semivolatile and intermediate-volatility organic compounds (S/IVOC) for different instrument inlet and tubing materials. The measured delay times could be explained by absorptive partitioning. The same model as developed earlier for VOCs could be applied for this data set by adjusting the material specific parameters. The results and the framework presented here is extremely useful for a proper design of instrument inlets and choice of tubing material to measure quantitatively low volatility multifunctional compounds. In a second part the authors also found that instrument response delay times for small polar molecules could be scaled with their Henry's Law coefficient. They partition to small amounts of water on the surfaces of the inlet or tubing. The manuscript is well written and data and results are clearly presented. The manuscript can be published as is. I have only a few minor comments.

R1.1. Line 227: you mean: residence time is 1 to 2 orders of magnitude shorter than diffusion time scale?

Yes. We changed "faster" to "**shorter**".

R1.2. Figure 5: DHC measurements are mentioned but not shown.

Revised the Figure 5 legend and caption to read:

"**DHCs were not included in delay calculation due to relatively low S/N ratios in these experiments. The transmission became lower than 100% for DHCs with C\* <$10^2$ μg m$^{-3}$ through steel and C\* <$10^3$ μg m$^{-3}$ through Silonite.**"

R1.3. Figure 6a: Could you indicate roughly the expected response time for these volatility classes; basically an extrapolation of Figure 2?

The instrument response time of these small, polar compounds will depend on absorption into small quantities of water on the instrument walls. So the response time cannot be extrapolated from the Figure 2 compounds, which exhibited a dependence mostly on volatility.
We have added the following text to the Figure 6 caption to partially address this point:

"**No delay would be expected for these compounds due to partitioning to Teflon, as they all have C\* > $10^7$ μg m$^{-3}$.**"

R2.0. Liu et al. present a detailed characterization of the signal delay in detecting organic vapors with saturation vapor concentration ($C*$) of $10^0 - 10^4 \mu g$ m$-3$ through different types of sampling tubes. Different types of CIMS have been used in this study to compare the effect of inlet design on signal delay in detection. RH effect is also probed. Adsorption or absorption of organic vapors by the tube wall under different situations are discussed. The characterization can be very helpful in designing an instrument's inlet for the detection of a fast-changing environment or quantification of gas-phase components. This manuscript is well-written and organized. I suggest for publication after considering the following aspects:

General comments about experimental suggestions:

R2.1. Though detailed suggestions have been given in Section 4, I am concerning some other points that the authors have not covered. For example, given the much slower desorption time of low volatile species (Fig. 1 and 4b), are we supposed to use new sampling tubes for every experiment?

We added some discussions in Section 4:

"**When accurate quantification in an environment with varying concentrations is necessary, the signals as a result of partitioning effects, if they cannot be eliminated, need to be separated from true ambient signals. Very recently, Palm and Thornton (2019) proposed a frequent, fast zeroing method to capture the HNO$_3$ signals due to memory effect in a CIMS IMR, immediately after the IMR volume has been cleared out and before the analyte can re-partition between the walls and the gas phase. This method can be potentially adapted to other compounds in other inlet and instrument configurations after careful examination.**"

R2.2. In Fig. 1, the instrument-only signal decay of the compound C10H21NO5 has not goes back to 0 in 2500 s. Does that mean the IMR has to be cleaned every time after detection of these species?

We repeated this type of experiment on a daily basis and did not see any noticeable residue signal due to IMR the next day, i.e. the background signal always went back to the same level after a sufficiently long time. Note that while the CIMS is not in use, it was always sampling clean air, which helped clean out what has been absorbed. We have added the following text to P7 L186 (as in submitted manuscript):

"**Note the long desorption timescales of the less volatile compounds.**"

If accurate measurement is needed in an environment that the compound concentration varies, please refer to our response to the response to R2.1.

R2.3. Humidifying IMR shortens the response time based on this study, but will that affect the ionization efficiency of the vapor molecules?

We had already included a related statement for the small, polar compounds (P19, L403 of the AMTD version): "Therefore when optimizing measurement response time for such small polar molecules, ambient humidity or water added on purpose (e.g., for enhancing CIMS sensitivities towards peroxyacyl nitrates, etc. (Slusher et al., 2004)) needs to be considered in addition to tubing and instrumental configurations."

Regarding the larger organic molecules, we have added the following text in the Instrument Response Section 3.1:

**"While humidifying an IMR can be beneficial for response time, the application of this method also needs consideration of water vapor's effect on ionization efficiency of different compounds (Lee et al., 2014). "**

R2.4. The other question is how to use the fact that low volatile species will level off after a while but with a low transmission efficiency, which could be an inverse problem for experiments without prior knowledge. I am sure the authors have the solution, but maybe a step-by-step process helps the readers a lot.

Unfortunately we do not have a universal solution to the problems created by this behavior. We added the following text at the end of Section 3.3 to add some recommendations of things to try:

**"If metal tubing cannot be avoided and an analyte's transmission is unknown, we recommend probing transmission efficiency first, such as by comparing the signals through different lengths of a same material and by investigating the linearity of signals vs. a range of analyte concentrations**."

R2.5. The last is how confident the authors are with the relationship between Cw and C∗. As the authors have shown the study of small polar molecules, how about the effect of functional groups of organic species? pH of the water film could play a role, how about the potential hydrolysis reactions?

Actually $C_w$ depends on the interaction between a class of compounds with a type of polymer material. The larger organic compounds are thought to be absorbed into the Teflon walls, and not interacting with the water film. We modeled $C_w$ in separate ranges of C* because irreversible loss was seen for lower C* compounds. However, for higher C* range ($10^2$-$10^4$ in this study), a single $C_w$ can be used to predict delay time for different C* values.

We already have a paragraph in Section 3.2 as well as Figure S3 that discuss that functional groups can affect $C_w$ through activity coefficient and that DHCs can cyclize to hydroxy cyclic hemiacetals on wet walls.

pH is expected to play a role for small molecules that partition mainly to the water films, and can dissociate. This was already addressed in the submitted manuscript, P18-19 L384-389. We have not modified the manuscript further to address this point.

Specific comments:

R2.6. Page 7 Line 172: Usually in the exponential fitting, the decay rate is k = 1/ τ =∑1/ τi. Though 10% is defined in this study, I would use the same expression.

The reason that we did not use this definition is that both instrument and tubing+instrument delays are not well captured by a single exponential fit, but they need to be fitted using 2 or 3 exponentials, which makes it complicated to determine the 10% response point from the fit equations. An example would be Figure 1, where the decays clearly have different τ values and different corresponding remaining signals. This is why we chose to instead use the empirically determined time to 10% of the original value as the metric to quantify this phenomenon. We have added the following text to P8 L 201 to clarify this point:

"The measured I-CIMS delay times (times to decrease to 10% of initial signals, **derived directly and without using the fitting results**) as a function of analyte C* are shown in Figure 2.

We have also added the word **"multiple"** before "exponential on P7 L186.

R2.7. Page 7 Line 186: About double and triple exponential fitting: what is the τ value reported in this paper?

See response to R2.6.

R2.8. Figure 2: Only several points have error bars. Do all of them have error bars or do other data points simply have small error bars?

The other points have similar error magnitudes, so we prefer not to make the plot too crowded by adding error bars to every single point. We have added the following text to the Figure 2 caption to clarify this:

"Error bars are shown **for some data points (the size would be similar for the rest of the points, not added to reduce figure clutter)**,..."

R2.9. Figure 4: The red curve (C6HN) higher than 1 is explained by competitive replacement by less volatile compounds, but how to explain the decrease? Though compounds with C* < 100 µg m$^{-3}$ have lower transmission efficiency, how to explain the fast response time (the overlap with species of higher C* at the beginning)?

For the first part of the question, we added: "**After ~ 8 min, the signals of C$_6$ HN dropped back to the Teflon-measured level, indicating no more extra desorption flux from the walls and that its adsorption and desorption had reached a steady state.**"

For the fast response time of these lower C* compounds, we added: "**Note that the initial responses of these lower-volatility compounds were as fast as the most volatile C$_6$ HN. This may be due to nearly irreversible losses of any molecule that had contact with the walls.**"

R2.10. Figure 5: Looks like Dihydroxycarbonyls are not in the figure.

See Response to R1.2.

Additional Change:

We have added the following paragraph after P14 L319. This is the same model used in the paper as submitted, and we are merely making it available to everyone and easy to use (upon requests from colleagues), without any changes to the results.

**"The model of Pagonis et al. (2017) has been revised to incorporate the $C_W$ parameters and tubing types tested in this work and in Deming et al. (2019), and also by adding a panel interface for ease of use. The updated model (v2), including the open source code, is included in the supplementary information of this paper. It is also available, together with instructions, at https://tinyurl.com/PartitioningDelays, where any future updates will also be posted."**